# Effect of Tertiary Lymphoid Structures on Prognosis of Patients with Hepatocellular Carcinoma and Preliminary Exploration of Its Formation Mechanism

**DOI:** 10.3390/cancers14205157

**Published:** 2022-10-21

**Authors:** Jianhui Li, Ye Nie, Weili Jia, Wenlong Wu, Wenjie Song, Yongxiang Li

**Affiliations:** 1Department of General Surgery, The First Affiliated Hospital of Anhui Medical University, Hefei 230022, China; 2Department of Hepatobiliary Surgery, Xijing Hospital, Fourth Military Medical University, Xi’an 710032, China

**Keywords:** tertiary lymphoid structures, hepatocellular carcinoma, LCK, immunotherapy, prognosis

## Abstract

**Simple Summary:**

At present, research on tertiary lymphoid structures (TLSs) in hepatocellular carcinoma (HCC) has been limited to the prognostic impact. Our manuscript first validates previous studies using two databases and then initially explores the key molecules and mechanisms of TLS formation and immunotherapy implications for HCC patients by using the TCGA database. For example, LCK, a key molecule in the formation of TLSs, may affect the formation of TLSs by regulating the cytokine signalling pathway, chemokine signalling pathway, T-cell activation and P53 signalling pathway. Second, the expression level of LCK is another factor affecting the sensitivity of HCC patients to immune checkpoint inhibitors. In conclusion, our study provides a potential mechanism for further exploration of TLSs.

**Abstract:**

Background: Tertiary lymphoid structures (TLSs) are formed by the aggregation of tumour-infiltrating lymphocytes (TILs), which is driven by chemokines or cytokines in the tumour microenvironment. Studies have shown that TLSs are associated with good prognosis in patients with various solid tumours and can improve patient responses to immunotherapy. However, the role of TLSs in hepatocellular carcinoma (HCC) remains controversial, and the underlying molecular mechanism is unclear. Methods: According to haematoxylin-eosin (HE) staining results, HCC patients in Xijing Hospital data and TCGA data were divided into TLS+ and TLS- groups, and Kaplan–Meier (KM) analysis was performed to assess overall survival (OS) and recurrence-free survival (RFS). Immunofluorescence (IF) and immunohistochemistry (IHC) were used to identify TILs in the TLS+ group. Lymphocyte-specific protein tyrosine kinase (LCK), a molecule involved in TLS formation, was explored in LinkedOmics. TILs were divided into two groups by drawing receiver operating characteristic (ROC) curves to calculate cut-off values. Spearman correlation analysis was used to calculate the correlation between LCK and TILs, and the molecular pathways by which LCK regulates immunotherapy were clarified through enrichment analysis. The half-maximal inhibitory concentration (IC50) distribution of sorafenib was observed in groups that varied in LCK expression. Results: According to the HE results, 61 cases in the Xijing Hospital cohort and 195 cases in the TCGA cohort had TLSs, while 89 cases and 136 cases did not. The KM results showed that TLSs had no effect on the OS of HCC patients but significantly affected RFS. The IF/IHC results showed that higher TIL numbers in TLSs were correlated with better prognosis in HCC patients. Spearman correlation analysis showed that LCK expression was positively correlated with TIL numbers. Enrichment analysis showed that upregulation of LCK expression mainly regulated the cytokine signalling pathway, the chemokine signalling pathway and T-cell activation. The IC50 scores of sorafenib in HCC patients with high LCK expression were lower, and the sensitivity was higher. Conclusion: TLSs mainly affected the early RFS of HCC patients but had no effect on OS. The high expression of the TLS formation-related gene LCK can increase the sensitivity of HCC patients to ICIs.

## 1. Introduction

In 2020, there were approximately 19.3 million new cancer cases worldwide, and liver cancer was the sixth most common and the third deadliest cancer [1]. Approximately 50% of new cases and deaths occur in China [2]. Hepatocellular carcinoma (HCC) is the most common clinical type of liver cancer, accounting for between 85 and 90% of cases [3]. In recent years, the diagnosis and treatment of HCC have greatly improved, but the overall resection rate of HCC is low, and the postoperative recurrence and mortality rates are high, resulting in a 5-year survival rate of approximately 15% among HCC patients [4,5]. The high recurrence and mortality rates of HCC are the most difficult problems to overcome at present. With the clinical application of neoadjuvant therapy, the resection rate of HCC has increased, and the survival period of patients with advanced HCC has been significantly prolonged [6]. However, the objective response rate (ORR) of HCC patients to immune checkpoint inhibitors (ICIs) and targeted drugs is low, which seriously limits the benefits of neoadjuvant therapy [7]. The tumour microenvironment (TME), which comprises tumour cells, tumour-infiltrating lymphocytes (TILs), stromal cells, etc., is highly heterogeneous and promotes both tumour growth and immune tolerance, eventually leading to the immune escape of tumour cells [8]. Therefore, TILs in the TME are considered to be the ultimate target of immunotherapy in HCC patients [9].

Tertiary lymphoid structures (TLSs) are newly discovered immune structures formed by the aggregation of lymphocytes in patients with chronic infections, immune diseases and tumours, which have the ability to initiate immune responses [10]. TLSs are mainly composed of CD3+ T cells, CD8+ T cells, CD20+ B cells, dendritic cells (DCs), macrophages and high endothelial venules, and can improve immune function and, thus, affect the prognosis of tumour patients [11,12]. TLSs can be divided into lymphocyte aggregates, round lymphocyte clusters without germinal centres, and follicles with germinal centres according to the degree of maturity [13]. Germinal centres are sites of B cell amplification, immunoglobulin class conversion, somatic hypermutation and affinity maturation, and the emergence of germinal centres can improve the prognosis of HCC patients [14,15]. Studies have shown that TLSs are associated with good prognosis in patients with gastric cancer [16,17], colorectal cancer [18,19], pancreatic cancer [20], lung cancer [11], bladder cancer [21], breast cancer [22], head and neck squamous cell carcinoma [23] and melanoma [24]. In addition, the ORRs of patients to targeted drugs and ICIs were better among tumour patients with TLSs than among tumour patients without TLSs [25]. However, the function of TLSs in HCC patients is still controversial. For example, Finkin et al. showed that para-tumour TLSs were associated with worse OS in HCC patients [26]. However, Calderaro and Li found that TLSs were not associated with the OS of HCC patients but rather were associated with better RFS of HCC patients [13,15], and the underlying mechanism remains unclear.

The mechanism underlying the anti-tumour effect of TLSs may be the result of a combination of many components, including T cell immunity, B cell humoral immunity, DC antigen presentation or the effect of a single component [27]. The formation and function of TLSs are dependent on the recruitment of lymphocytes by CCL2, CCL-3, CCL-4, CCL-5, CCL-8, CCL-18, CCL-19, CCL-21, CXCL9, CXCL-10, CXCL-11 and CXCL-13 [28,29]. Lymphocyte-specific protein tyrosine kinase (LCK) is a member of the Src family of protein tyrosine kinases [30]. LCK is a key signalling molecule that functions in T cell maturation, mediates T cell metastasis and enhances the anti-tumour effects of T cells in response to stimulation by chemokines [31,32,33]. In patients with nasopharyngeal carcinoma, LCK promotes tumour proliferation and increases drug resistance [31]. However, it remains unknown whether LCK mediates the formation of TLSs and, thus, affects the molecular mechanism underlying the efficacy of immunotherapy and targeted therapy in HCC patients. Therefore, we studied 150 HCC patients from Xijing Hospital and public databases to investigate the mechanism of action of TLSs and LCK in regulating HCC responses to immunotherapy.

## 2. Material and Methods

### 2.1. Patients

In this study, HCC patients who underwent surgical resection in the Department of Hepatobiliary Surgery of Xijing Hospital from 2014 to 2018 were numbered, and 150 HCC patients were finally selected by random sampling through SPSS software; all HCC patients received surgical resection for the first time without any other anti-tumour therapy. The liver function, alpha-fetoprotein (AFP) level, tumour number, tumour diameter and pathological stage of HCC patients were assessed in the study. Patients were followed up by telephone every 3 months from the day of surgery to ask whether the patient’s condition had changed. The follow-up time was up to 30 June 2020. The median OS and RFS of HCC patients in this cohort were 33.1 and 17.3 months, respectively. This study was approved by the Ethics Committee of Xijing Hospital (KY20202115-C-1) and was conducted in accordance with the criteria of the Declaration of Helsinki.

### 2.2. Real-Time Quantitative Fluorescence PCR

Frozen cancer tissues and adjacent tissues of 39 HCC patients were removed from liquid nitrogen, and approximately 150 mg of frozen tissue was cut into EP tubes containing 1 mL Trizol (Accurate Biology, https://agbio.com.cn/, accessed on 15 October 2021, AG21102) and ground in a tissue grinder. Total RNA was extracted from tissues according to the instructions of the RNA extraction kit SYBR Premix EX Taq^TM^ II (Accurate Biology, AG11702). Finally, the extracted RNA was reverse transcribed according to the kit instructions (Accurate Biology, AG11706). A maximum of 500 ng Total RNA was used in the SYBR Green qPCR assay in a 10 μL reverse transcription system. qRT–PCR was used to detect the expression levels of LCK and other genes in HCC tissues, and β-actin was used as a control to calculate the relative mRNA expression levels of genes by the 2^−ΔΔCt^ method.

All primer sequences were designed and synthesised by Tsingke Biological (tsingke.com.cn, accessed on 28 September 2021), and primer sequences used for amplification are shown in Table 1.

### 2.3. Haematoxylin-Eosin (HE) Staining

HE staining was used to analyse formalin-fixed paraffin-embedded HCC tissue samples from Xijing Hospital according to the kit instructions (Servicebio, https://www.servicebio.cn/, accessed on 13 November 2021, G1003). HE staining results were evaluated by two senior pathologists from Xijing Hospital. The patients were divided into the TLS+ group and the TLS-group according to the presence or absence of TLSs observed in the HE results.

### 2.4. Immunofluorescence (IF) and Immunohistochemistry (IHC) Staining

IF and IHC were performed in the TLS+ group. Anti-CD20 antibody (Abcam, ab78237, 100-fold dilution), anti-CD3 antibody (ab135372, 100-fold dilution), anti-CD8 antibody (ab237709, 100-fold dilution) and anti-LAMP3 antibody (ab271053, 100-fold dilution) were added to the refrigerator overnight at 4 °C, and the corresponding secondary antibody (100-fold dilution) was added the next day. The nuclei were counterstained with DAPI (Servicebio, G1012) and then photographed. The sum of CD20+ B cells, CD3+ T cells, CD8+ T cells and LAMP3+ DCs in all TLSs in a 400× field was calculated manually. The numbers of positive cells were counted and converted to cell density (cells/mm^2^) [15,17]. The TILs were divided into high- and low-count groups by drawing receiver operating characteristic (ROC) curves to calculate the cut-off value ((sensitivity + specificity) − 1).

### 2.5. Downloading and Screening Data

The sequencing data and corresponding clinical data of HCC patients were downloaded from The Cancer Genome Atlas (TCGA) database. Excluding 46 patients who received anti-tumour therapy (except surgical resection), 331 patients were included in the study.

### 2.6. Screening of TLS-Related Molecules

We used LinkedOmics to download the relevant molecular data for the chemokines (CCL2, CCL-3, CCL-4, CCL-5, CCL-8, CCL-18, CCL-19, CCL-21, CXCL9, CXCL-10, CXCL-11 and CXCL-13). The molecules related to all 12 chemokines were selected, and LCK, whose expression was the most highly correlated with TIL numbers and was highest in HCC, was selected using the GEPIA and TIMER websites. Then, we used Xijing Hospital data to calculate the correlations between LCK expression and TILs numbers. In addition, TCGA data in the ACLBI (Assistant for Clinical Bioinformatics, https://www.aclbi.com/static/index.html#/, accessed on 3 August 2021) database were used to calculate the immune score of the different LCK expression groups to evaluate the impact of LCK expression on the response to immunotherapy.

### 2.7. Enrichment Analysis

To study the molecular mechanisms by which LCK expression affects TLS formation and immunotherapy response in HCC patients, TCGA data were used to perform Gene Ontology (GO) and Kyoto Encyclopedia of Genes and Genomes (KEGG) analyses to explore the relevant molecular pathways. *p* < 0.05 (namely, −log10 > 1.3) was considered statistically significant.

### 2.8. Half-Maximal Inhibitory Concentration (IC50) Values

The IC50 is an important indicator for evaluating drug efficacy or sample response to treatment. Genomics of Drug Sensitivity in Cancer (GDSC) data were used to calculate differences in HCC patients’ sensitivity to sorafenib in the different LCK subgroups.

### 2.9. Survival Analysis

For all the above groups (TLS+ and TLS- groups, high and low TILs count groups and high and low LCK expression groups), Kaplan–Meier (KM) survival analyses were performed. The log-rank test was used to compare survival differences among different groups, and Cox analysis was used to determine the independent prognostic factors of HCC patients. *p* < 0.05 was considered statistically significant.

### 2.10. Statistical Analysis

Unpaired t tests were used to calculate significant differences in LCK expression between the TLS+ and TLS- groups with GraphPad Prism 6 Software. Cox analysis was used to determine the independent factors that affected the RFS of HCC patients. Survival differences were expressed by a KM curve and analysed by log-rank tests. Spearman correlation analyses were also performed. Related R packages included “ggplot2”, “stats”, “pheatmap”, “clusterProfiler” and “pRRophetic”. *p* < 0.05 was considered statistically significant.

## 3. Results

### 3.1. Intratumoural TLS

According to the HE results, among the 150 HCC patients from Xijing Hospital, 61 patients (40.7%) were assigned to the TLS+ group and 89 patients (59.3%) were assigned to the TLS-group (Figure 1). In addition, 195 patients in the TCGA cohort had TLSs and 136 did not (Appendix A). The KM survival curve results showed that the presence of TLSs had no effect on the OS of HCC patients (Figure 2A); however, the presence of TLSs was associated with good RFS (Figure 2B), especially early RFS (within 2 years after HCC resection [32]) (Figure 2C), but had no effect on advanced RFS (Figure 2D). Verification of the above results using TCGA data yielded the same results (Appendix A). Univariate Cox analysis showed that the presence of TLSs, aspartate transaminase (AST), lymph, alkaline phosphatase (ALP), albumin (ALB), tumour diameter, tumour capsule, cancer emboli, Child grade, American Joint Committee on Cancer (AJCC) stage and Barcelona Clinic Liver Cancer (BCLC) stage were significantly correlated with RFS. Multivariate Cox analysis showed that the presence of TLSs, AST, lymph, ALB, tumour capsule, Child grade and BCLC stage were independent prognostic factors of RFS (Table 2). TCGA data also verified that TLSs were an independent protective factor for RFS in HCC patients (Appendix A). Therefore, the presence of TLSs, as an independent protective factor for RFS, may be associated with longer RFS in HCC patients. The clinicopathological characteristics of 150 HCC patients in Xijing hospital are shown in Appendix A.

### 3.2. TILs of Intra-TLS

To further show the protective effect of TLSs in HCC patients, the numbers of CD20+ B cells, CD3+ T cells, CD8+ T cells and LAMP3+ DCs in TLSs were observed by IF/IHC (Figure 3). According to the cut-off values, CD20+ B cells, CD8+ T cells, CD3+ T cells and LAMP3+ DCs were divided into high expression or low expression groups. The results showed that the numbers of CD20+ B cells and CD8+ T cells were significantly correlated with better OS in HCC patients (Figure 4A,B), while the numbers of CD3+ T cells and LAMP3+ DCs were not significantly correlated with OS (Figure 4C,D). However, both numbers were significantly correlated with the RFS of HCC patients (Figure 4E–H). In other words, the more CD20+ B cells, CD8+ T cells, CD3+ T cells and LAMP3+ DCs in TLSs, the better the prognosis of HCC patients. Moreover, the numbers of these cell populations had a strong predictive validity for the RFS of HCC patients (Figure 5). The AUCs of CD20+ B cells, CD3+ T cells, CD8+ T cells and LAMP3+ DCs were 0.678, 0.663, 0.716 and 0.802, respectively. LAMP3+ DCs have the largest AUC, which may provide guidance for post-operative survival prediction and clinical treatment of HCC patients.

### 3.3. Molecular Mechanisms by Which TLSs Regulate the Response to Immunotherapy

#### 3.3.1. TLS-Related Molecule Expression

The expression of a total of 88 molecules was positively correlated with the expression of 12 chemokines, with a correlation greater than 0.4 (moderate correlation) (Figure 6A). LCK, whose expression was most highly correlated with TILs numbers and was highest in HCC, was selected. Spearman correlation analysis was used to calculate the correlation between the expression of LCK and chemokines in Xijing Hospital, TCGA and ICGC data (Figure 6B–D). In addition, the expression of LCK in the TLS+ group was significantly higher than that in the TLS-group (Figure 7A). According to the cut-off value of the LCK mRNA expression level, LCK was divided into high and low expression groups. The results showed that the OS of HCC patients with a high expression of LCK was significantly better than that of the low expression group (Figure 7B).

#### 3.3.2. Immunotherapy

To test the effect of LCK expression on the response to immunotherapy, TIMER was used to prove that LCK expression was positively correlated with the numbers of B cells, T cells, DCs, macrophages and neutrophils, and the Xijing Hospital data were used to verify the results (Figure 8). In addition, our study showed that memory B cells, M1 macrophages, memory CD4+ T cells, CD8+ T cells and follicular helper T (Tfh) cells in the high LCK expression group were significantly higher than those in the low LCK expression group (Figure 9). M2 macrophages and resting dendritic cells were significantly lower than those in the low expression group (Figure 9). Moreover, the expression of LCK was positively correlated with the expression of PD-1, PD-L1 and CTLA4 (Figure 10A–C). However, some studies have shown that the high expression of PD-1/PD-L1 inhibits the sensitivity of HCC patients to ICIs [33], and some studies have shown that the high expression of PD-1/PD-L1 can be used as a benchmark for evaluating HCC immunotherapy [34]. Therefore, to further prove the influence of LCK expression on the response to immunotherapy, we used the MCP-counter algorithm to generate an immune score heatmap based on the differences in LCK expression in the TCGA data (Figure 10D,E); the results showed that the immune score of the group with high LCK expression was significantly higher than that of the group with low LCK expression, *p* < 0.05 (the higher the immune score, the higher the sensitivity of tumour patients to drugs). Second, by plotting the correlation between LCK and the P53 signalling pathway, we found that LCK expression was positively correlated with P53 expression (Figure 10F), further proving that LCK may increase the sensitivity of HCC patients to ICIs.

#### 3.3.3. Enrichment Analysis

To elucidate the mechanisms by which LCK regulates TLS formation and the response to immunotherapy in HCC patients, we performed KEGG and GO enrichment analyses. As shown in the figure, upregulation of LCK expression significantly affected cytokine–cytokine receptor interactions, the chemokine signalling pathway, T cell activation and other related molecular pathways (Figure 11A,B). Downregulation of LCK expression mainly affected cellular metabolism and other related molecular pathways (Figure 11C,D). Therefore, LCK may play an important role in the formation of TLSs.

#### 3.3.4. IC50 Scores

Currently, targeted therapy for HCC is at a bottleneck stage, mainly due to the immune escape of tumour cells and the immune tolerance of HCC patients to targeted drugs. The IC50 scores of 331 GDSC data was used to evaluate the sensitivity of HCC patients to sorafenib in different LCK groups. Our study showed that the IC50 values of sorafenib in the high LCK expression group were significantly lower than those in the low LCK expression group (Figure 12). Therefore, TLSs and LCK expression can be used as potential targets for precision treatment of individual HCC patients.

## 4. Discussion

In recent years, precision therapy for HCC, such as neoadjuvant therapy, has made significant breakthroughs in clinical practice, not only improving the resection rate of HCC but also significantly extending the median survival of HCC patients [35]. However, the ORRs of HCC patients to targeted drugs and ICIs have remained low, seriously limiting the clinical application of neoadjuvant therapy [36]. Studies have shown that the high expression of immune checkpoints on TILs inhibits the anti-tumour responses of T and B cells, so TILs may be the ultimate target of immunotherapy [37]. TLSs are recently discovered immune structures that are formed by lymphocyte aggregation that can enhance the ORRs of tumour patients to targeted drugs and ICIs [24]. However, the formation of TLSs is dependent on the recruitment of T and B cells by chemokines, so the regulation of chemokine production is key to the anti-tumour function of TLSs [16]. Studies have shown that CCL19 and CCL21 mediate T cell and DC migration and that CXCL13 activates B cells, which are critical conditions for TLS formation [29]. In addition, Tfh cells are necessary for B cells to exert humoral immunity; one of the reasons that B cells accumulate in the TME is the secretion of CXCL13 by Tfh cells [38]. B cells can also play the role of antigen presenting cells, secreting anti-inflammatory cytokines to further enhance the anti-tumour ability of T cells, DCs and NK cells [39]. B cells have been shown to function as lymphoid tissue inducer cells, secreting IL-22 to regulate TLS formation [40]. TLSs, in turn, promote tumour antigen presentation, T cell activation and B cell antibody secretion, thus, triggering the development of adaptive anti-tumour immune responses [41]. This illustrates the capacity of TLSs to exert effects beyond those on B cells and participate in the activation of the adaptive immune system in a local immune response [29]. Our results demonstrate that the expression of LCK is positively correlated with that of chemokines, that LCK is a key signalling molecule in T cell maturation, and that chemokines can further enhance T cell metastasis and anti-tumour activity by stimulating LCK expression [30,42,43]. The enrichment analysis showed that LCK expression could positively regulate or activate cytokine–cytokine receptor interactions, the chemokine signalling pathway and the T cell activation signalling pathway. CXC chemokines are mainly involved in cytokine–cytokine receptor interactions [44]. In addition, previous studies have shown that LCK can increase the expression and migration of chemokines (such as CCL19, CCL21, CXCL9, CXCL10, CXCL11 and CXCL13) [45,46]. Therefore, due to this positive feedback regulation, LCK expression may be a factor that participates in the formation of TLSs.

The TLS-related molecule CXCL13 is associated with good prognosis in HCC patients [47], and the expression of CCL19 and CCL21 in tumours can promote the anti-tumour effect of TILs [48]. However, chemokines released by HCC (CXCL-2, CXCL8 and CCL25), in turn, increase the proportion of neutrophils in the TME, forming an immunosuppressive microenvironment and leading to tumour progression [49,50]. According to our results, LCK expression is positively correlated with the number of neutrophils, and increased expression of LCK promotes the aggregation of neutrophils in the TME. Neutrophils, however, are associated with a worse prognosis in HCC patients and are resistant to anti-tumour immunotherapy in HCC patients [51,52]. This contradictory phenomenon does not negate the protective effects of LCK in HCC. This is because tumour-associated neutrophils (TANs) can be divided into anti-tumorigenic (N1) and pro-tumorigenic (N2). Among them, N1-TAN can recruit and activate immune cells by producing cytokines, chemokines and proteases, which can stimulate T cell proliferation and NK cell and DC maturation [53,54,55,56]. For example, N1-TAN can promote the recruitment and activation of CD8+ T cells by producing TNF-α, CCL3, CXCL9 and CXCL10 [57]. Therefore, whether LCK may promote the formation of TLSs and increase the anti-tumour function of TLSs through N1-TAN induction of CCL3, CXCL9 and CXCL10 aggregation remains to be further investigated. In summary, the complexity and functional diversity of TME components do not depend on the function of a single component.

PD-1 and PD-L1 are expressed not only by T cells and tumour cells but also by macrophages, DCs, B cells and other immune cells. Therefore, ICIs can indirectly activate TLSs by blocking the PD-1/PD-L1 pathway and thereby activate the expression of immune cells, thus, enhancing the anti-tumour effect [58]. For example, nivolumab and pembrolizumab treat advanced liver cancer by reducing the expression of PD-1 and thereby decreasing T cell depletion [33]. In addition, high expression of PD-L1 can increase the level of AFP in the blood of HCC patients, thus, promoting tumour progression [59]. Interestingly, previous studies have shown that elevated PD-L1 expression in HCC supports the development of immune checkpoint inhibitors targeting the PD-1/PD-L1 pathway, and the expression levels of CD8+ T cells and PD-L1 can be used as benchmarks to evaluate new therapies in HCC [34]. These therapies initiate tumour immune checkpoint suppression therapy by inducing TILs and increasing PD-L1 expression in HCC [34]. In addition, elevated immune checkpoint expression of PD-1 and PD-L1 was demonstrated in the immune microenvironment of MSI-H colorectal cancer (CRC), and a phase II study of pembrolizumab in CRC patients showed an increased rate in immune-related objective response in MSI-H-CRC compared with MSS-CRC [60]. To verify that LCK may promote the sensitivity of HCC patients to ICIs, we calculated the immune scores of HCC patients in different groups based on differences in LCK expression. The results showed that HCC patients with high expression of LCK had higher immune scores. The higher the immune score is, the higher the objective response rate of HCC patients to ICIs [61,62]. Activation and maturation of T cells can increase the sensitivity of HCC patients to ICIs [63,64]. When LCK activates T cells, the secretion of IL-2 increases, which, in turn, activates CD8+ T cells within the tumour to increase the ability of ICIs to inhibit tumours and reduce drug toxicity [65,66]. Second, by studying the correlation between LCK and the P53 signalling pathway, we found that LCK expression was positively correlated with P53 expression, and previous studies found that P53 could increase the sensitivity of HCC patients to ICIs [67,68]. Third, we studied the differences in the IC50 distribution of sorafenib based on differences in LCK expression. The results showed that the IC50 values of sorafenib in HCC patients with high LCK expression were significantly lower than those in HCC patients with low LCK expression, indicating that HCC patients with high LCK expression were more sensitive to sorafenib. Sorafenib combined with ICIs was significantly better than sorafenib monotherapy in treating patients with advanced HCC (e.g., the combination improved median survival and progression-free survival) [69]. In conclusion, LCK may increase the sensitivity of HCC patients to ICIs. Although our study suggests that LCK may be a key molecule affecting TLSs, our results provide a promising entry point for the specific molecular mechanism of TLS formation and the reason why TLSs increase the sensitivity of HCC patients to ICIs and improve the prognosis of patients.

## 5. Conclusions

Our study indicates that TLSs are an independent protective factor for RFS in HCC patients. Upregulation of LCK expression may further promote the formation of TLSs and improve the sensitivity of HCC patients to targeted drugs and ICIs. Therefore, TLSs and LCK are potential therapeutic targets.

## Figures and Tables

**Figure 1 cancers-14-05157-f001:**
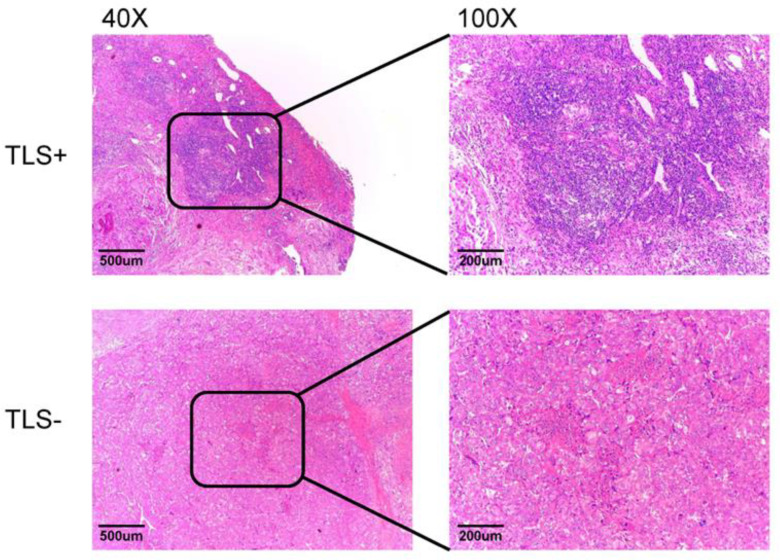
According to the results of HE staining, HCC patients in Xijing Hospital were divided into TLS+ (*n* = 61) and TLS- groups (*n* = 89).

**Figure 2 cancers-14-05157-f002:**
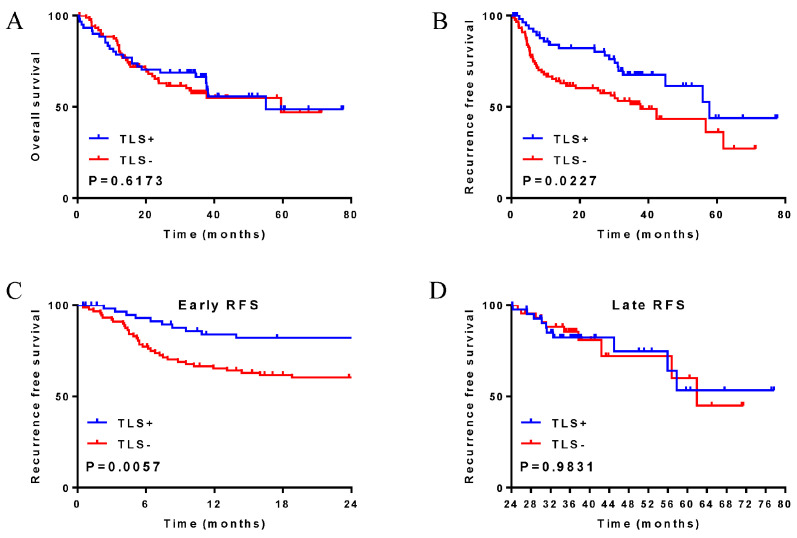
Xijing Hospital data, effects of TLSs on overall survival and recurrence-free survival in patients with hepatocellular carcinoma. (**A**) overall survival; (**B**) recurrence-free survival; (**C**) early recurrence-free survival; (**D**) late recurrence-free survival.

**Figure 3 cancers-14-05157-f003:**
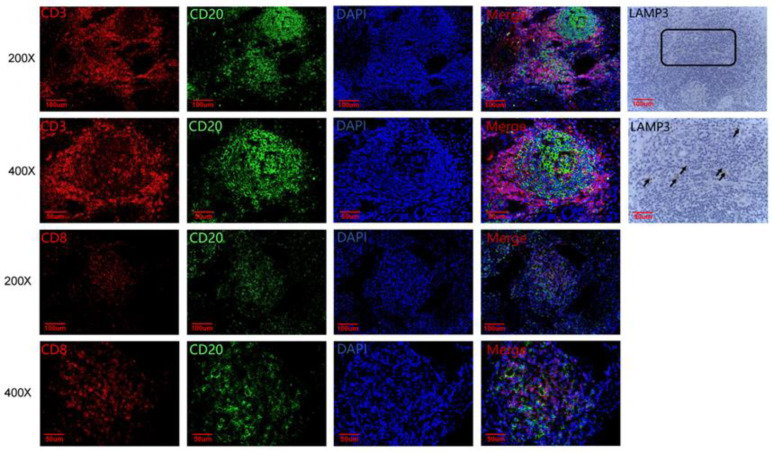
Xijing Hospital data, immunofluorescence/immunohistochemistry staining of CD20+ B cells, CD3+ T cells, CD8+ T cells and LAMP3+ DCs in TLSs.

**Figure 4 cancers-14-05157-f004:**
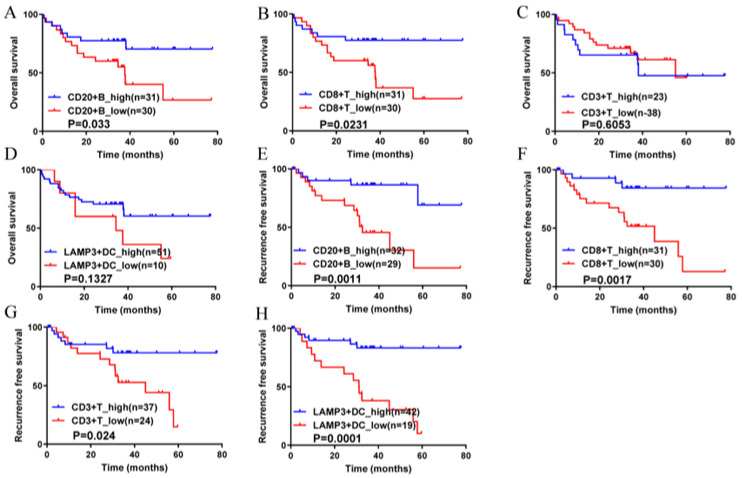
Xijing Hospital data, effects of CD20+ B cells, CD3+ T cells, CD8+ T cells and LAMP3+ DCs in TLSs on overall survival and recurrence-free survival in patients with hepatocellular carcinoma. (**A**) The higher the expression of CD20+B cells, the better the OS of HCC patients; (**B**) The higher expression of CD8+ T cells, the better OS of HCC patients; (**C**) CD3+ T cells have no effect on OS of HCC patients; (**D**) The higher expression of LAMP3+DC cells, the better OS of HCC patients; (**E**–**H**) The higher the expression of CD20+B cells, CD8+ T cells, CD3+ T cells and LAMP3+DC cells, the better the RFS of HCC patients. CD20+ B cells, CD3+ T cells, CD8+ T cells and LAMP3+ DCs were divided into a high expression group and a low expression group according to the cut-off value.

**Figure 5 cancers-14-05157-f005:**
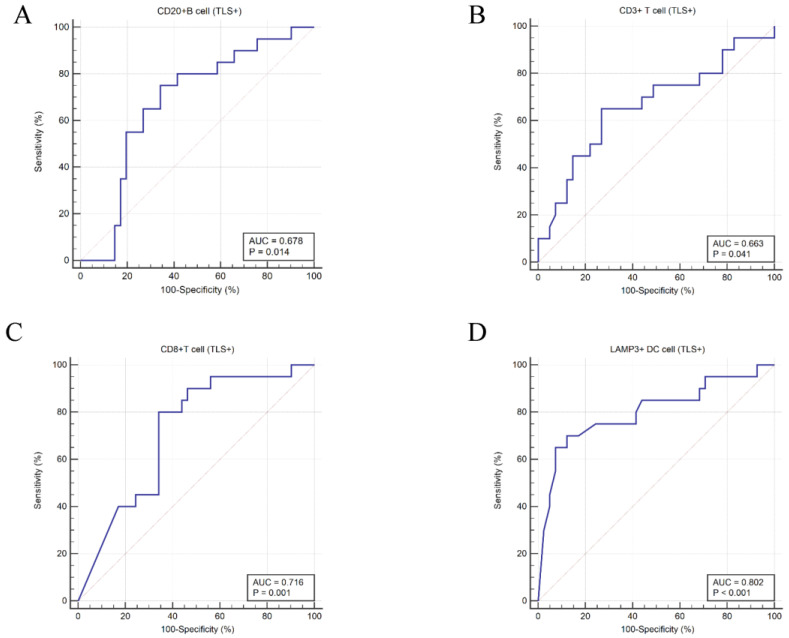
Xijing Hospital data, ROC curves drawn based on the levels of TILs in TLSs. (**A**) CD20+ B cells; (**B**) CD3+ T cells; (**C**) CD8+ T cells; (**D**) LAMP3+ DCs.

**Figure 6 cancers-14-05157-f006:**
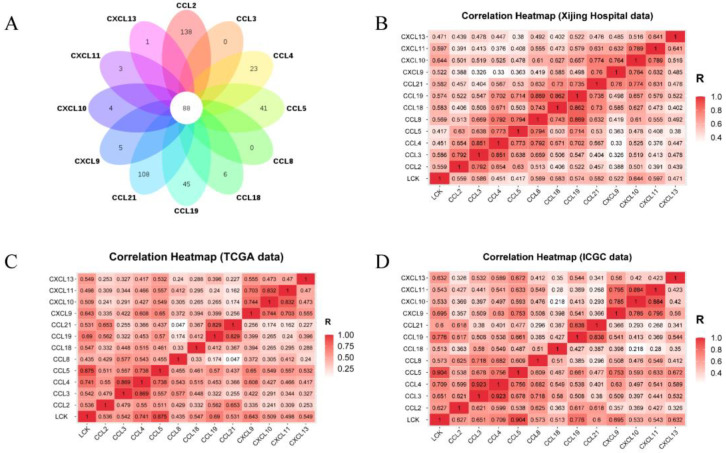
(**A**) Venn diagram of 12 chemokine-related molecules predicted with LinkedOmics. Spearman correlation heatmaps of LCK and chemokine expression from (**B**) Xijing Hospital data; (**C**) TCGA data and (**D**) ICGC data.

**Figure 7 cancers-14-05157-f007:**
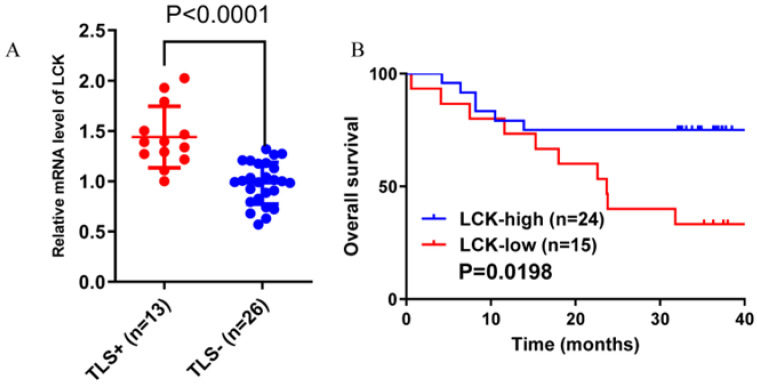
Xijing Hospital data. (**A**) The expression of LCK in the TLS+ group was significantly higher than that in the TLS-group. (**B**) The overall survival of patients in the high LCK expression group was significantly higher than that of patients in the low LCK expression group.

**Figure 8 cancers-14-05157-f008:**
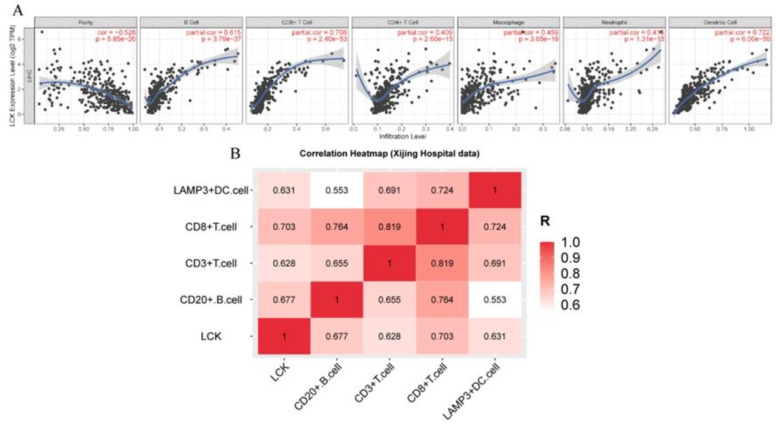
Correlation analysis of LCK expression and levels of tumour-infiltrating lymphocytes: (**A**) TIMER data and (**B**) Xijing Hospital data.

**Figure 9 cancers-14-05157-f009:**
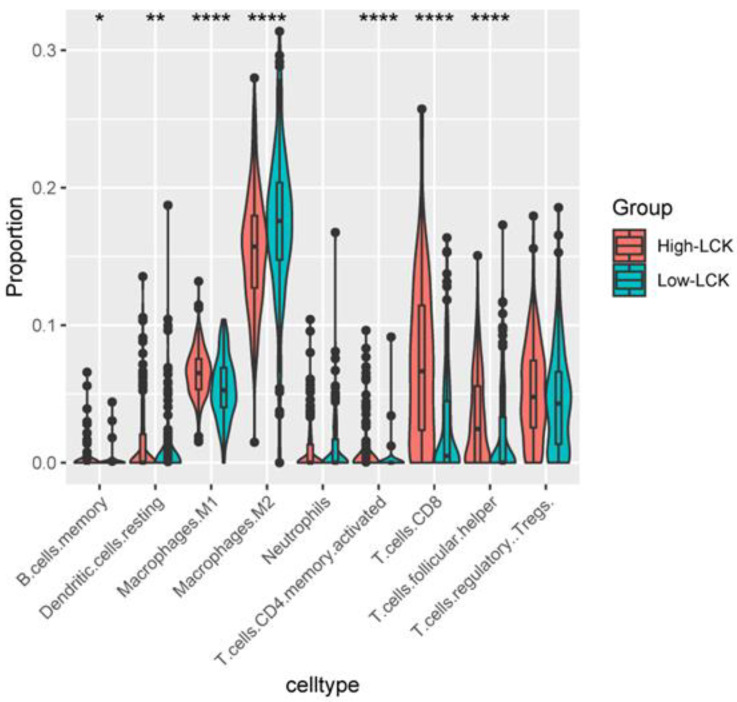
TCGA data, CIBERSOFT was used to calculate the infiltration of immune cells in each sample, and the R package ggplot2 was used for visualisation. The rank sum test was used to explore the differences in TILs between the high and low LCK expression groups. *p* < 0.0001 = ****, *p* < 0.01 = **, and *p* < 0.05 = *.

**Figure 10 cancers-14-05157-f010:**
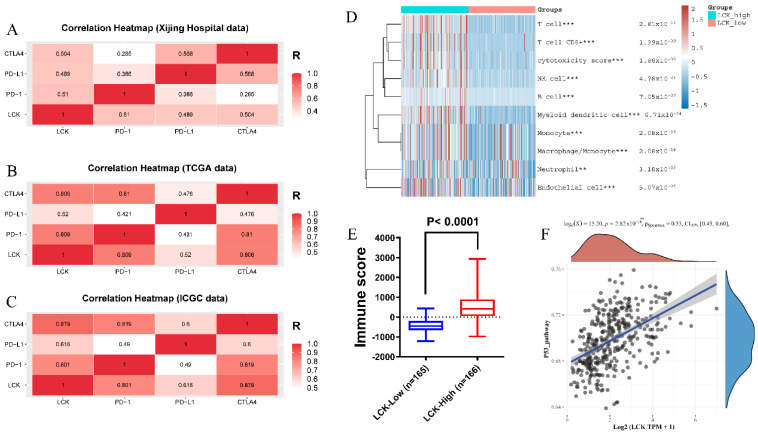
Correlation heatmaps of LCK expression and immune checkpoint expression from the Spearman correlation analysis: (**A**) Xijing Hospital data; (**B**) TCGA data and (**C**) ICGC data. (**D**) Using R software to draw the immune scoring heatmap based on the expression difference in LCK and MCP counter algorithm; (**E**) Box chart of immune score difference between different LCK expression groups. (**F**) Scatter plot of the correlation between LCK and the P53 signalling pathway. *p* < 0.001 = *** and *p* < 0.01 = **.

**Figure 11 cancers-14-05157-f011:**
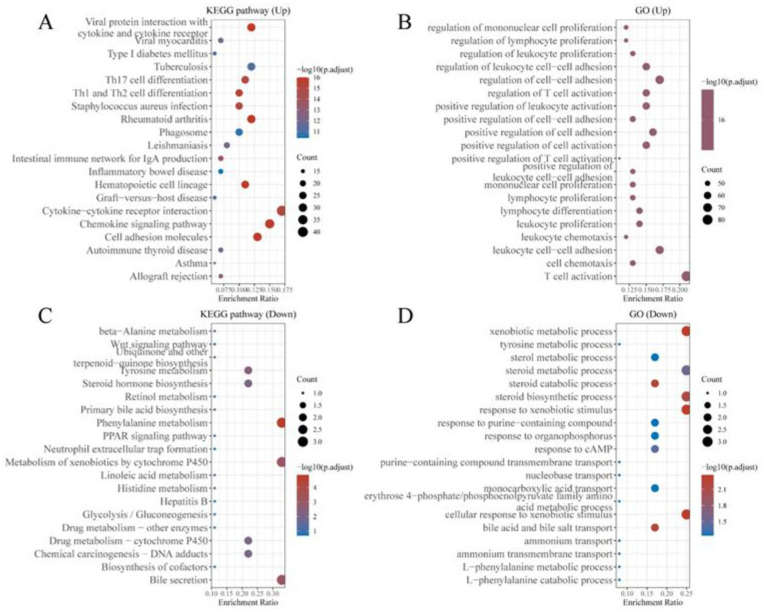
Enrichment results of LCK regulatory functions calculated using TCGA data: (**A**) KEGG pathways upregulated by LCK; (**B**) GO pathways upregulated by LCK; (**C**) KEGG pathways downregulated by LCK; (**D**) GO pathways downregulated by LCK.

**Figure 12 cancers-14-05157-f012:**
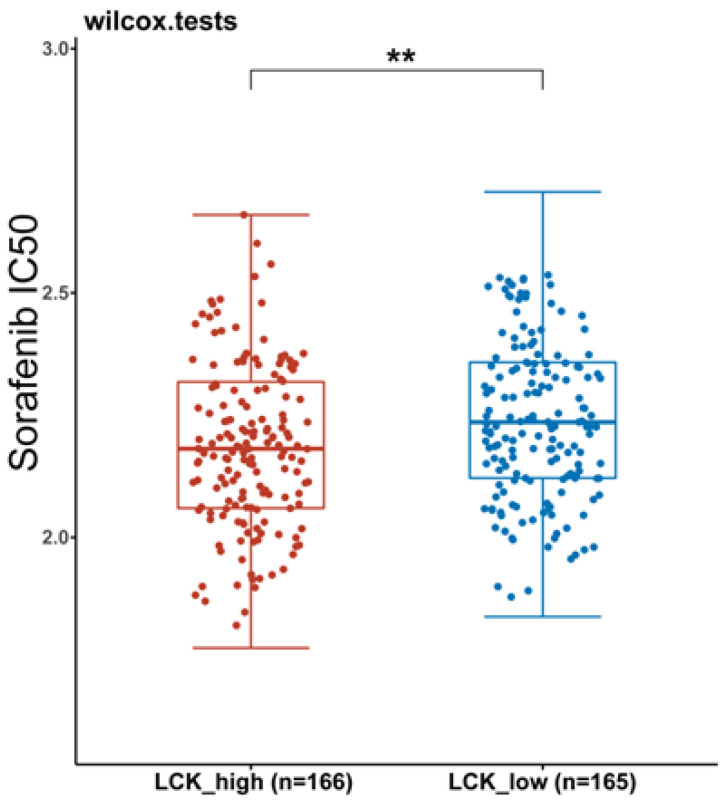
Distribution of the half-maximal inhibitory concentrations of sorafenib in the two LCK expression groups. *p* < 0.01 = **.

**Table 1 cancers-14-05157-t001:** Primer sequences used in quantitative real-time polymerase chain reaction (qRT–PCR).

Gene	Forward (5′-3′)	Reverse (5′-3′)	Accession Numbers
LCK	CACGAAGGTGGCGGTGAAGA	GAAGGGGTCTTGAGAAAATCCA	009P2021112200886 009P2021112200890
CCL2	GCTCATAGCAGCCACCTCATTC	CCGCCAAAATAACCGATGTGATAC	009P2021082300911 009P2021082300925
CCL3	ATCATGAAGGTCTCCACCAC	TCTCAGGCATTCAGTTCCAG	009P2021082300912 009P2021082300926
CCL4	TGCTAGTAGCTGCCTTCTGC	TTCACTGGGATCAGCACAGAC	009P2021082300927 009P2021082300913
CCL5	CCAGCAGTCGTCTTTGTCAC	CTCTGGGTTGGCACACACTT	009P2021082300914 009P2021082300928
CCL8	TGGAGAGCTACACAAGAATCACC	TGGTCC AGATGCTTCATGGAA	009P2021082300915 009P2021082300929
CCL18	CTCTGCTGCCTCGTCTATACCT	CTTGGTTAGGAGGATGACACCT	009P2021082300916 009P2021082300930
CCL19	CTGCCTGTCTGTGACCCAGCGCCCC	ACTTCTTCAGTCTTCGGATGATGCG	009P2021082300917 009P2021082300931
CCL21	CCTTATCCTGGTTCTGGCCT	CAGCCTAAGCTTGGTTCCTG	009P2021082300918 009P2021082300932
CXCL9	ATGAGGATGAAAGTGGTGATTGG	GGTGTTGGTGTTGAATAGAAAGC	009P2021082300919 009P2021082300933
CXCL10	ATGAGGATGAAAGTGGTGATTGG	GGTGTTGGTGTTGAATAGAAAGC	009P2021082300920 009P2021082300934
CXCL11	GACGCTGTCTTTGCATAGGC	GGATTTAGGCATCGTTGTCCTTT	009P2021082300921 009P2021082300935
CXCL13	GCTTGAGGTGTAGATGTGTCC	CCCACGGGGCAAGATTTGAA	009P2021082300922 009P2021082300936
PD-L1	TGGCATTTGCTGAACGCATTT	TGCAGCCAGGTCTAATTGTTTT	009P2021080500650 009P2021080500651
PD-1	CCAGCCCCTGAAGGAGGA	GCCCATTCCGCTAGGAAAGA	009P2021081000727 009P2021081000728
CTLA4	GCCCTGCACTCTCCTGTTTTT	GGTTGCCGCACAGACTTCA	009P2021080900751 009P2021080900752
β-actin	CTCCATCCTGGCCTCGCTGT	GCTGTCACCTTCACCGTTCC	009P2021051700660 009P2021051700661

**Table 2 cancers-14-05157-t002:** Univariate and multivariate Cox regression analyses of risk factors associated with recurrence-free survival from the Xijing Hospital data (HCC patients did not receive any other anti-tumour therapy prior to tumour resection).

Variable	Univariate Analysis		Multivariate Analysis	
	HR (95% CI)	*p* Value	HR (95% CI)	*p* Value
Sex (male vs. female)	1.258 (0.621–2.547)	0.524		
Age (>60 vs. ≤60)	1.005 (0.587–1.720)	0.987		
TLSs (positive vs. negative)	0.545 (0.321–0.926)	0.025	0.504 (0.291–0.875)	0.015
ALT, µ/L (>50 vs. ≤50)	1.038 (0.627–1.720)	0.885		
AST, µ/L (>40 vs. ≤40)	1.974 (1.206–3.232)	0.007	1.874 (1.125–3.122)	0.016
HBsAg (positive vs. negative)	1.437 (0.802–2.577)	0.223		
TBil, µmol/L (>20.5 vs. ≤20.5)	1.033 (0.604–1.786)	0.905		
AFP, ng/mL (>400 vs. ≤400)	1.504 (0.913–2.478)	0.109		
Liver cirrhosis (Yes vs. No)	1.179 (0.713–1.952)	0.521		
Lymph, 10E9/L (>1.1 vs. ≤1.1)	0.535 (0.319–0.899)	0.018	0.521 (0.299–0.909)	0.022
ALP, µ/L (>125 vs. ≤125)	2.879 (1.741–4.760)	<0.0001		
ALB, g/L (>40 vs. ≤40)	0.345 (0.207–0.576)	<0.0001	0.382 (0.225–0.649)	<0.0001
Tumour number (multiple vs. single)	1.743 (0.927–3.278)	0.085		
Tumour diameter, cm (>5 vs. ≤5)	2.369 (1.435–3.911)	0.001		
Tumour capsule (Yes vs. No)	0.442 (0.270–0.724)	0.001	0.333 (0.198–0.562)	<0.0001
Cell differentiation (healthy vs. poor/moderate)	0.623 (0.349–1.112)	0.109		
Portal vein invasion (Yes vs. No)	1.889 (0.857–4.165)	0.115		
Cancer emboli (Yes vs. No)	1.810 (1.065–3.075)	0.028		
Child grade (B/C vs. A)	4.608 (1.664–12.760)	0.003	4.266 (1.465–12.421)	0.008
AJCC Stage (III + IV vs. I + II)	2.384 (1.435–3.962)	0.001		
BCLC Stage (B + C vs. 0 + A)	2.123 (1.289–3.495)	0.003	2.466 (1.458–4.170)	0.001

Notes: TLS, tertiary lymphoid structures; ALT, alanine aminotransferase; AST, aspartate transaminase; TBil, total bilirubin; AFP, alpha-fetoprotein; ALP, alkaline phosphatase; ALB, albumin.

## Data Availability

The datasets generated and/or analysed during the current study are available from the TCGA and ICGC repositories.

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
