# Peer review of "Effect of Tertiary Lymphoid Structures on Prognosis of Patients with Hepatocellular Carcinoma and Preliminary Exploration of Its Formation Mechanism"

_cancers, 2022, doi:10.3390/cancers14205157_

Round 1
Reviewer 1 Report
The authors showed that the formation of tertiary lymphoid structures(TLS) in the HCC microenvironment was associated with a great prognosis for HCC patients and enhanced recurrence-free survival, which had been reported in the Journal of Hepatology (PMID: 30213589). Nevertheless, they also showed that LCK might be associated with the formation of TLS, which could be a potential therapeutic target for HCC with TLS, resulting in enhanced efficiency of immune checkpoint inhibitor or targeted drug. Their experiments are well controlled in general. Here are some comments:
1. It is better to display the included quantity of each group on each figure and legend. Moreover, the norm of classification of the two groups should be mentioned. For example, how to determine the high expression of CD8, CD3, CD20 and LAMP3 in the TLS+ group in figure 4? And how to determine the LCK-high group in Figure 7, whether it is according to the mean relative expression level of mRNA of LCK?
2. To better verify the correlation between LCK and the formation of TLS, it’s recommended that immunofluorescent double staining of LCK and CD20 or CD3 be monitored.
3. Given that PD-1 blockade or other ICI treatment would impact the tumor immune microenvironment, it’s better to mention that whether any treatment was chosen before HCC patients received resection or not in the table.
4. When showing the expression of CD8 in Figure 3 (line 3), it’s suggested that CD20 or CD3 be marked simultaneously to verify TLS formation.
5. The differentiation and infiltration of regulatory T cells in TME is also associated with CCL5/CCR5 signaling, which was the main factor of failure of immunotherapy. Thus, it’s recommended that Foxp3+ Treg be considered in the formation of TLS and LCK expression.
Author Response
Dear reviewer,
Please see the attachment.
Kind regards,
Yongxiang Li

Reviewer 2 Report
The manuscript of Li et al. reports a study on the correlation between the presence of Tertiary Lymphoid Structures (TLSs) and the prognosis of patients with HepatoCellular Carcinoma (HCC) using different databases and a cohort from surgical department of Xijing Hospital. The link between Lymphocyte-specific protein tyrosine Kinase (LCK) and the sensitivity of patients to immune checkpoint inhibitors (ICIs) was also tested. Using mainly statistical analyses, the authors concluded that TLSs are independent protective factors for Recurrent-free Survival in HCC patients and propose a molecular mechanism linked to LCK expression and action leading to improved sensibility to ICIs.
General comments:
Given the incidence and mortality of HCC and the low Objective Response Rate (ORR) of patients to ICIs, the data obtained are potentially of great interest. Moreover, owing to the recent description of TLSs, the main topic of this study is in fact attractive. Thus, this statistical study based on a substantial amount of work provide important additional information on the consideration of intratumoral TLSs and on the potential role of LCK as a marker or potential target for HCC.
The manuscript is globally well written. However, in the present form, the manuscript presents some weaknesses to aspire to be published in Cancers in the present form.
Regarding the form of manuscript, the referee is worried about the lack of precision in the material and methods section and in the legend of figures. Important information is missing to consider some results:
- RT-qPCR: Lack of information for kit use, calculation method … Accession numbers for primers and/or references must be given.
- IF/IHC: How many field were counted to manually calculate the TILs number in patient specimen. Are isolated TILs present in TLS- group?
- In general, figure legends are not explicit enough and information is missing to fully exploit the results i.e.: Figure 2: Are TCGA data included or not?
- Results are rarely described and interpreted in the text. i.e.: Figure 3: A comparative description of infiltrating cells could be done; Figure 5: AUC value is clearly higher for dendritic cells which could have a better predictive validity.
Although statistical analyses were globally well conducted, some data are not sufficiently discernible and interpretable at first glance. (i.e.: Figure 6B and 9D). Statistical analysis in figure 11 is not fully convincing and the number of experiments/collected data should be mentioned (idem in figure 7A).
In addition, contradictory results with other studies appear on the number of neutrophils and on the expression of PD-1/PD-L1 related to the expression of LCK and have only been refuted based on a positive correlation of immunoscores. The descriptive approach could be completed by in vitro experiments performed with HCC cell lines to validate the role of LCK on these phenomena. No perspective of the study are proposed.
Finally, in conclusion section, the authors mentioned (lines 390-391) that “Our study …and elucidates the molecular mechanisms by which TLSs regulate the HCC response to immunotherapy". In my opinion, the term "elucidated" is exaggerated given that only a statistical correlation was shown without demonstration of a direct causal link.
Minor points:
- As mentioned in the Cancer Genome Atlas website, all database must be mentioned in the acknowledgements section. An example of a proper acknowledgement is:
"The results <published or shown> here are in whole or part based upon data generated by the TCGA Research Network: https://www.cancer.gov/tcga."
- Not really exploited in the result section, table 3 could be transferred to supplementary material.
- Line76: “canters”
Author Response

(The authors gave the same response as above.)
